# Categorization and Analysis of Relevant Factors for Optimal Locations in Onshore and Offshore Wind Power Plants: A Taxonomic Review

**Isabel C. Gil-García** [1,†] , **M. Socorro García-Cascales** [2,†] , **Ana Fernández-Guillamón** [3,†] and **Angel Molina-García** [3,*,†]

1   Faculty of Engineering, Distance University of Madrid (UDIMA), c/ Coruña, km 38.500 28400, Collado Villalba, 28029 Madrid, Spain; isabelcristina.gil@udima.es

2   Department of Electronics, Computer Architecture and Projects Engineering, Universidad Politécnica de Cartagena, 30202 Cartagena, Spain; socorro.garcia@upct.es

3   Department of Electrical Engineering, Universidad Politécnica de Cartagena, 30202 Cartagena, Spain; ana.fernandez@upct.es

*   Correspondence: angel.molina@upct.es; Tel.: +34-968-32-5462

†   These authors contributed equally to this work.

**Abstract:** Wind power is widely considered to be a qualified renewable, clean, ecological and inexhaustible resource that is becoming a leader in the current energy transition process. It is a mature technology solution that was quickly developed and has been massively integrated into power systems in recent years. Indeed, a remarkable number of renewable integration policies have been promoted by different governments and countries. With the aim of maximizing the power given by wind resources, the locations of both onshore and offshore wind power plants must be optimized following a sort of different criteria. Under this scenario, a number of factors and decision criteria in the evaluation and selection of locations can be identified. Moreover, the relevant wind power increasing in the power generation mix is addressed, along with a standardization of factors and decision criteria in the optimization and selection of such optimal locations. In this context, this paper describes a systematic review and meta-analysis combining most of the contributions and studies proposed during the last decade. Thus, our aim is focused on reviewing and categorizing all factors to be considered for optimal location estimation, pointing out the differences among the selected factors and the decision criteria for onshore and offshore wind power plants. In addition, our review also includes an analysis of the representative key indicators for the contributions, such as the annual frequency of publications, geographical classification, analysis by category, evaluation method and determining factors.

**Keywords:** wind energy; optimal selection factors; onshore-offshore wind power plant

## 1. Introduction

The depletion of fossil fuels, combined with the need to reduce greenhouse gas emissions, leads to a single result: energy transition [1,2]. Renewable energies, both inexhaustible and clean sources, constitute the main support for a sustainable energy transition [3]. The year 2017 culminated records for renewable energy systems installed. There was the largest increase in global capacity, reaching 2180 GW of total capacity, of which 53% belonged to hydraulic energy and 24% to wind energy (onshore + offshore) [4]. The wind industry stands out for its exponential growth during the last decade, both in accumulated MW and generated energy (See Figure 1). During 2017, 13 European and American countries reached 10% or more of their electricity consumption with wind energy, and according to

the IEA Wind sources, the global generated energy amounted to 1430 T Wh. Offshore wind energy reached its best year ever, with an increase of 67% compared to 2016.

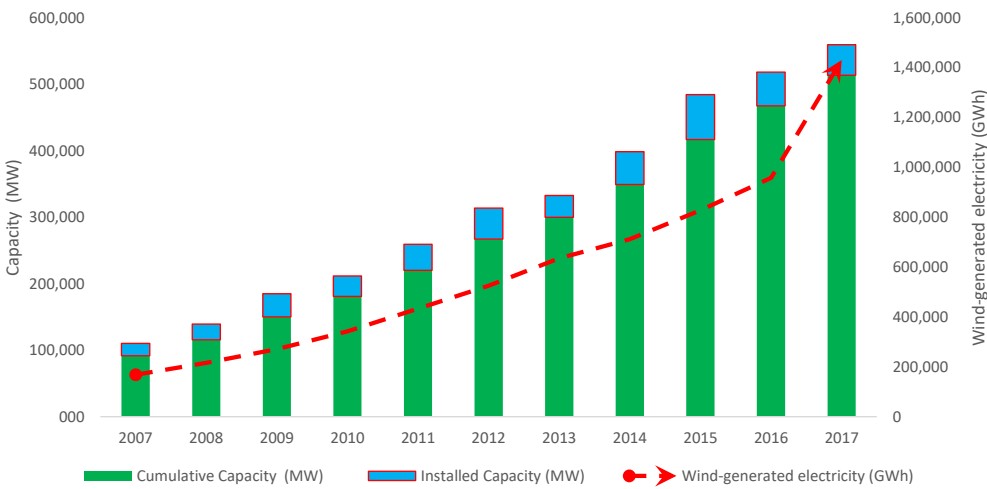

**Figure 1.** Comparison of accumulated, installed and generated wind energy. Data source: [5].

A significant cost reduction has been supported by an increasingly mature framework: gradual technological innovations, considerable improvements in the supply chain, reduction of the risk premium, greater qualification of developers and operators and a large market volume. Although the current wind market trend is increasingly favorable and optimistic, there are relevant barriers to overcome, mainly related to the optimal geographic location of wind power plants—both onshore and offshore. In this way, inaccurate forecasts of wind energy production can lead to the inefficiency of the wind facilities and, subsequently, to large financial losses. Presently, researchers and organizations work with the purpose of developing new optimization methodologies in the selection and evaluation of wind sites. According to the specific literature, the location of the wind power plants is associated with a group of factors that guarantee the profitability of such installations. However, and despite the extensive literature available, there is a lack of literature reviews dealing with the relational identification of the determining factors of such generation units for both onshore and offshore wind power plants. Under this framework, this paper gives an extended analysis and revision of the factors and determining decision criteria to select the optimal location of wind power plants for both onshore and offshore solutions. In addition, a categorization proposal of such factors is also provided by the authors according to the typology of the different factors. The works reviews in this paper constitute a clear evidence of the non-existence of a categorization of factors and criteria to be used for the efficient evaluations of onshore and offshore wind locations. The present analysis thus contributes to the literature by categorizing the factors and criteria—in terms of relevance, which influence the evaluation and selection of optimal locations for onshore and offshore wind plants.

The rest of the paper is structured as follows: Section 2 describes the research methodology used in the study, Section 3 presents the results and discussion on onshore wind facilities and offshore facilities with a comparison of both, and finally, Section 4 provides the conclusions and future work.

## 2. Proposed Methodology

The general proposed methodology of this review process is based on the Preferred Reporting Items for Systematic Reviews and Meta-Analyses (PRISMA) statement [6]. The main objective of this PRISMA statement is to address researchers in the selection of systematic review articles, guaranteeing the quality of the process. Many previous studies of different research categories have used the PRISMA statement to collect an exhaustive literature review [7–11]. In our case, the general

proposed methodology based on the PRISMA statement has two main processes: systematic review and meta-analysis. Figure 2 summarizes the proposed methodology.

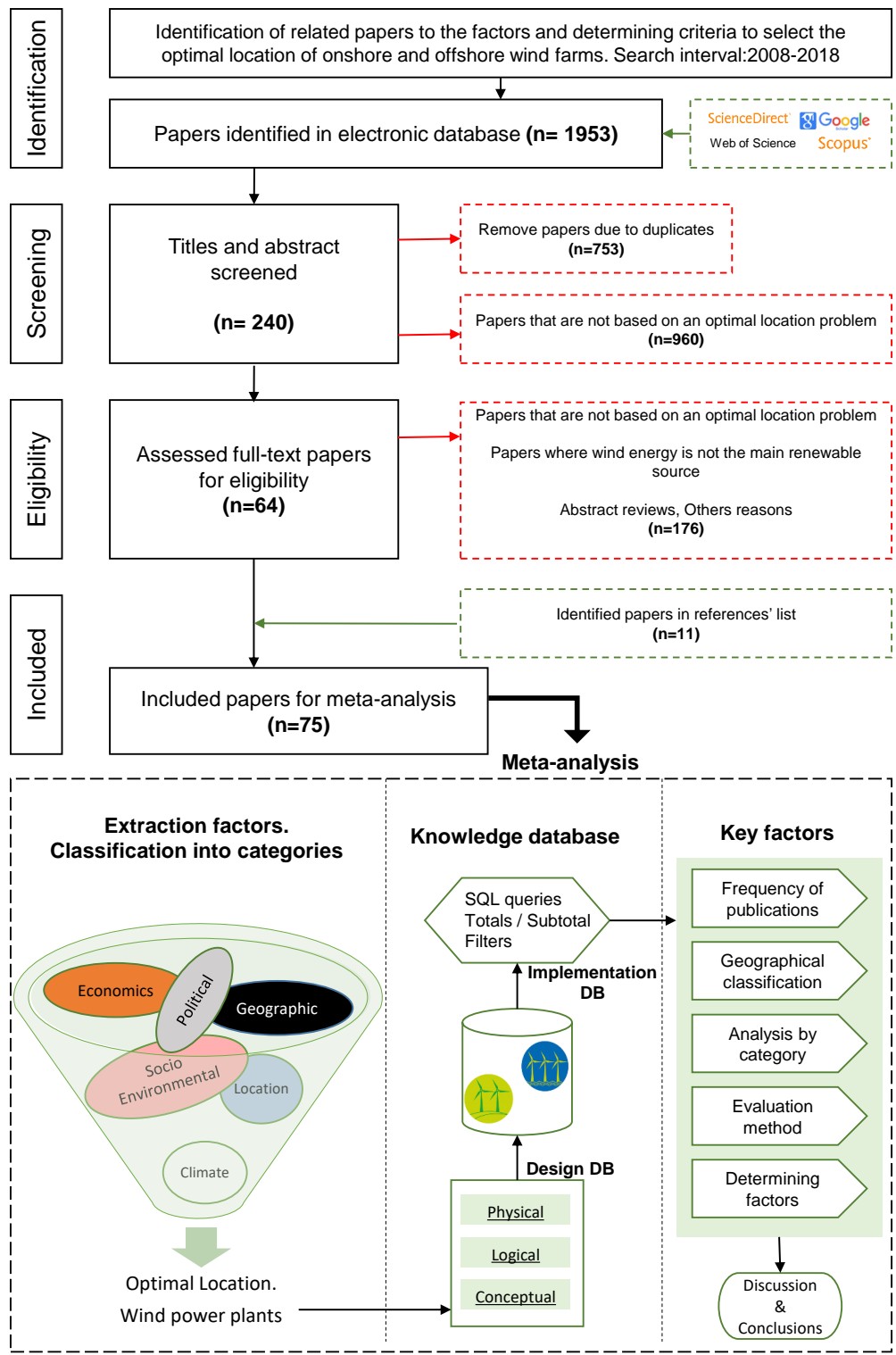

**Figure 2.** Proposed methodology: systematic review and meta-analysis.

*2.1. Systematic Review*

From the objective previously described in Section 1, this literature review aims to provide an extended analysis of the factors and determining criteria to select the optimal location of onshore and

offshore wind power plants. Studies corresponding to the last decade (2008–2018) were selected for this study, in line with the evolution and exponential growth of installed wind power capacity and technological maturity during those years. The systematic review can be then divided into four stages: Identification, Screening, Eligibility and Included—see Figure 2.

According to the different contribution databases currently available, the authors included the following: ScienceDirect, Google Scholar, Web of Science and SCOPUS. Based on the keywords of the general objective and the objectives derived from the analyzed topic, a total of 1953 records were identified. Subsequently, from the contributions related to the specific problem of optimal location of power plants from the title and summary fields, we identified 753 duplicated items and 960 that were not works related to the research objectives. Finally, 240 potential contributions were identified. Once the objective of eligibility was applied, we reviewed the full text of each work, discarding revisions by the information given by the abstracts. Therefore, papers where wind resource was not the main renewable source and those that did not have the optimal location problem as their main objective were identified. In this process, 176 works were rejected, and six additional works from the reference lists were included in the total works to be considered by this analysis. Finally, 75 contributions were selected to be studied.

*2.2. Meta-Analysis*

Three main steps were identified in this process: extraction factors, knowledge database and key factors. First, all the factors involved in the optimal location of the specific sites were extracted. The real meaning of each factor was studied and then, we proposed a categorization based on the studies carried out by the different authors, regardless of the case study. This solution constitutes a contribution to this important barrier—'Use of the wind resources'—of the wind industry [12]. Moreover, it can be considered to be a reference for future works by providing identification and categorization factors summarized in Tables 1–6.

**Table 1.** Description of factors involved in the optimal location in the wind farms. Climate category.

| Factor | Description |
|---|---|
| Wind speed | The wind speed that measures its kinetic energy in the site (m/s) |
| Power density | Power density, consider wind speed and air density (W/m$^2$) |
| Wind direction | Side where the wind blows (sexagesimal degrees) |
| Effective time | Occurrence of wind speed |
| Availability data | Accurate measurement campaign data |
| Turbulence | Ratio between the standard deviation of the values wind speed and its average speed, for each set of ten-minute measurements (dimensionless) |
| Frost periods | Duration of frost periods |
| Natural disasters | Probability of natural disasters |
| Air density | Relationship between mass and air volume (kg/m$^3$). Influences the kinetic energy of the air |

**Table 2.** Description of factors involved in the optimal location in the Wind farms. Geographic category.

| Factor | Description |
|---|---|
| Slope | The higher the percentage of the slope of the land, the less likely it is to install the wind farm |
| Altitude | At higher altitude, installation difficulties increase |
| Type of terrain | Soft or hard consistency |
| Roughness | Roughness of the terrain caused by both natural elevation and human development |
| Area | Area contained within the perimeter of the wind farm (m$^2$) or limit of the external ocean, legal marine areas of the country |
| Water depth | Bathymetry. Water depth in selected area of the sea (m). It is a key technical factor to decide the type of structure (fixed or floating) |
| Wave height | Wave height in selected area of the sea (m). It is a key technical factor to determine the effects of waves on the structure (balancing, dragging) |
| Water quality | It includes some properties of water such as dissolved oxygen (mg/L) to exclude areas destined for aquaculture or study co-location |

**Table 3.** Description of factors involved in the optimal location in the wind farms. Socio-environmental category.

| Factor | Description |
| --- | --- |
| Protected areas or distance | Completely protected areas from a legal standpoint (National and natural parks, Integral and special Natural Reserves, Special Areas of Conservation, etc.) |
| Agrological capacity | Suitability of the soil for certain crops |
| Visual impact | Visual impact according to regulations |
| Reduction emissions $CO_2$ and others | Pollution avoided compared to conventional power generation technology |
| Stroboscopic effect | Blinking shadow effect caused by the sun's incidence on the blades of the wind turbine |
| Energy-dependence contribution | Energy savings |
| Noise | The noise impact in quality of life |
| Population | The level and regularity of demand for energy in the site |
| Demand electricity | Sufficient electricity demand that justifies the installation |
| Land use | Use of land for agricultural, governmental, etc. Purposes |
| Flora and fauna impact | Mainly influence in birds, marine species, soil and vegetation |
| Shipping Routes | Ships/vessels movement routes |
| Fishing areas | Areas determine by the authorities for fishing |

**Table 4.** Description of factors involved in the optimal location in the wind farms. Location category.

| Factor | Description |
| --- | --- |
| Distance/Availability roads | Distance to roads, focused on decreasing installation and maintenance costs as well as safety in everyday transport |
| Distance to other wind farms | With the purpose of not exceeding the estimation of the carrying capacity of sustainable siting areas |
| Distance transmission lines (antennas) | Distance between any telecommunications infrastructure and the wind farm. In order to not affect the telecommunications infrastructure |
| Distant urban areas | Distance between urban areas, towns or cities, and location areas. In anticipation of future expansions and in compliance with the legislative framework of any country |
| Distances industrial/Military zones | Distance between military and industrial zones and location areas |
| Distance from the railway network | Distance between railway lines and possible locations. With the aim of taking advantage of the social acceptance of the zones |
| Distances to ports | Distance between ports and the possible sites, adaptation to the country's regulatory framework |
| Distances airports | Distance between the nearest airport and the different possible sites with the objective of not affecting the airspace or the future expansion of airports and facilities. Airspace restricted by the Aviation Agency |
| Distance to Point of Common Coupling (PCC) | Distance between nearest network or power line and the different possible sites. While this distance is smaller, the cost of the electricity infrastructure is lower and therefore, the economic and financial indicators will be better |
| Distances entertainment areas—historical | Distance between entertainment, historical areas and the possible sites, adaptation to the country's regulatory framework |
| Distance water resources (rivers, coast, lake) | Distance between water resources and the possible sites, adaptation to the country's regulatory framework, depending on whether it is a lake, river etc. |
| Distance of underground cables or pipes | Distance or existence of underground cables or pipes |
| Distance to shore | Focused on the location of offshore wind farms by regulatory measures marked by the country |
| Distance other point | Distance to other point as wrecks, lighthouses |

**Table 5.** Description of factors involved in the optimal location in the wind farms. Economic category.

| Factor | Description |
| --- | --- |
| Energy sale price | Energy sale price, very important since it is the only source of income for the installation |
| Energy put into the network | Energy put into the network eliminated all losses of gross energy |
| Infrastructure cost | Costs of the infrastructure associated with the initial investment (CAPEX) |
| NPV | Net present value, financial indicator |
| IRR | Internal rate of return, financial indicator |
| Payback | Recovery period in years |
| Interest loan | Interest of the loan requested in the initial investment |
| Installed capacity | Installed capacity (MW) |
| Exploitation | Cost focused on the exploitation phase (OPEX), example: cost of land (onshore), port activities (offshore) |
| Stability voltage | Voltage stability to achieve the planned energy |
| Economic contribution | Economic contribution focused on the creation of employment, payment of taxes in town halls etc. |
| Decommission cost | Include the removal of the turbines and foundations (DECEX) |

**Table 6.** Description of factors involved in the optimal location in the wind farms. Political category.

| Factor | Description |
|---|---|
| Incentives | Incentives received in compensation for producing electric power from renewable sources |
| Taxes | Taxes involved in the activity |
| Policy measures | Political measures established in favor of renewable energies |

From the initial categorization process, we designed a proposed knowledge database according to the requirements and objectives of the different contributions. The conceptual model of the database was then designed, translating the entities and the relationships between them. After that, the logical design was determined, normalizing the database to avoid duplication of information. In the implementation of the database, each contribution was inserted and proceeded to program queries that respond to our objectives: filters, totals, subtotals, groupings, etc.

Finally, the key factors can be estimated by considering the following aspects:

- Frequency of publications: a first analysis of the frequency of annual publications is analyzed to identify the period in which these studies became more relevant.
- Geographical classification: to identify the geographical areas with the greatest impact of publications and their possible association with indicators from different fields (governmental measures in favor of renewable energy, social acceptance, etc.) a study is carried out by country, marine area and continent.
- Quantitative analysis: to quantitatively analyze the categories and their associated factors, it is calculated by each contribution the following aspects: the number of times such factors are used in the contributions of each technology (onshore and offshore), and their percentage of use with respect to those contributions.
- Evaluation method: in the process of searching for and selecting such optimal locations for wind power plants, it is possible to identify (i) a large amount of spatial information, and (ii) the need to cluster factors and criteria from varied nature which influence with different intensities in the multicriteria decision-making. Many researchers who tried to address the complexity of these investigations have proposed to use Geographic Information System (GIS) tools and/or Multicriteria Decision-Making (MCDM) methods. Given the importance of the methodological development of these contributions, a third indicator can be identified focused on analyzing the percentage of the researchers providing a methodology that combines geographical information systems and MCDM, or they use any of them individually.
- Determining factors: they are based on the previous analysis. The first ten most relevant determining factors are identified for each onshore and offshore technology.

## 3. Results and Discussion

### 3.1. Onshore Analysis. Categorization and Factors

In the taxonomic review of the contributions focused on the optimal selection of onshore wind power plant locations, 34 relevant works published between 2008 and 2018 were selected. From these studies, we can affirm that from 2008 to 2013, the frequency of publications was considerably low in comparison to the rest of the period. Indeed, only 10 contributions aiming to optimize onshore wind power plants were found in the specific literature until 2013 (29.4% of the total works). However, in the period 2014–2018, 24 works were published. Figure 3 shows the frequency of publications for onshore wind power plants classified by the author's origin. Therefore, and according to this contribution review, the selection of an optimal location indicator became more attractive in the early 2010s. In line with the previous results, the number of publications according to the country of the case study was obtained accordingly. Thus, the countries that have published the highest number of case studies are Spain (5), the USA (4), Greece, Iran, Turkey (3), Brazil, and China (2). Clustering these contributions to provide a continental ranking, Asia leads the list with 38%, followed by Europe with 35%, North America with 12%, South America with 9% and Africa with 6%. Figure 4 shows the geographical

classification of case studies for onshore wind power plant locations (2008–2018). These results conclude that studies of evaluation and selection of optimal onshore installation locations are mostly driven by developed countries during the last decade. From the categorization and qualitative analysis described in Section 2.2, both factor identification and categorization is following described for onshore wind power plant optimal location.

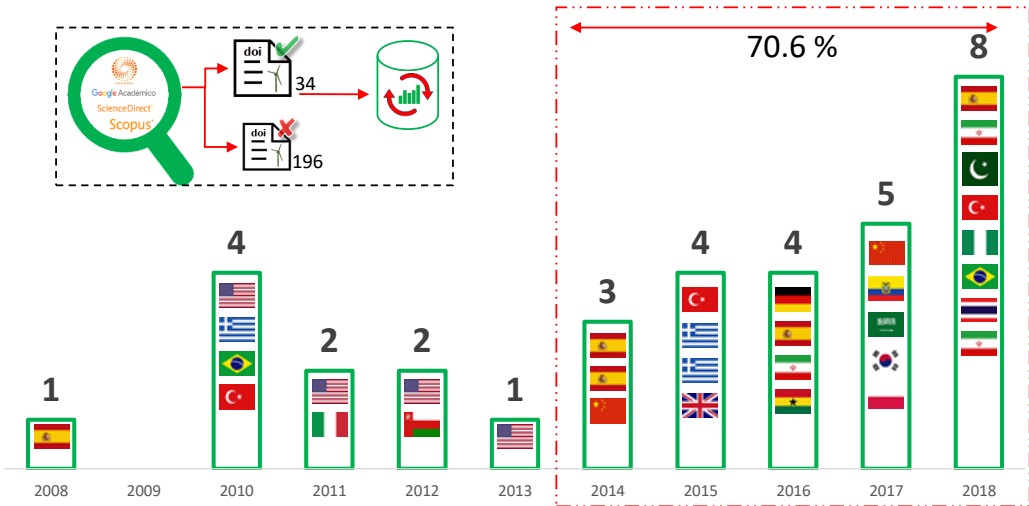

**Figure 3.** Onshore wind power plant optimal location. Frequency of publications (2008–2018).

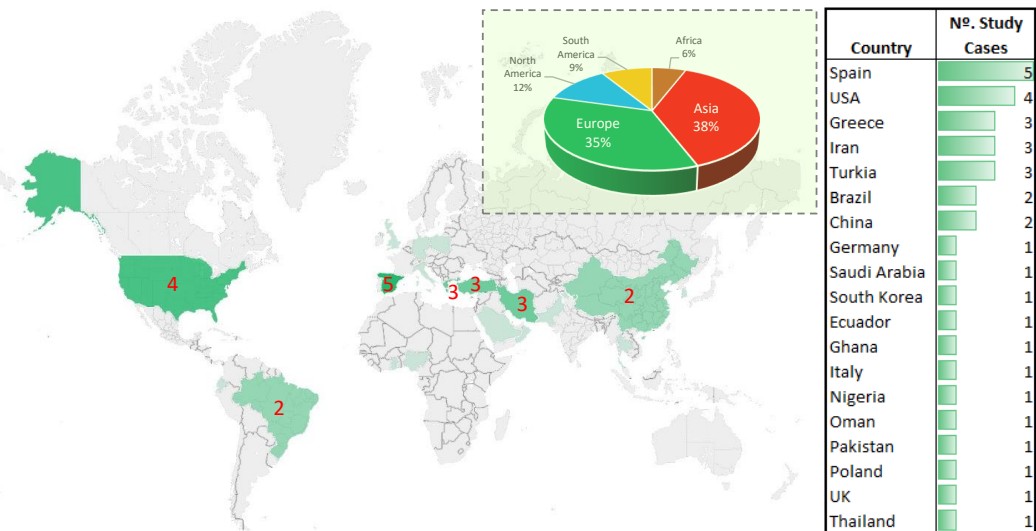

**Figure 4.** Onshore wind power plant optimal location. Geographical case studies (2008–2018).

### 3.1.1. Climate Category ($C_1$)

In line with Table 1, a total of nine factors were included in this category, labeled from ($C_{1.1}$) to ($C_{1.9}$). Table 7 summarizes the main contributions by including (or not) the factors corresponding to the climate category for onshore optimal location methodologies. Both references and the number of works—labeled as Absolute Frequency (AF)—are included in the table, as well as the percentage of contributions where such factor is considered in the different studies. According to the results, the most relevant factor is Wind Speed ($C_{1.1}$), accounting for 32 works that directly included it in their studies for optimal location; followed by Air Density ($C_{1.9}$) with five contributions. The least relevant factors were Data availability ($C_{1.5}$), Turbulence ($C_{1.6}$) and Frost periods ($C_{1.7}$), with only one study.

**Table 7.** Climate Category ($C_1$)—Onshore optimal location. Quantitative analysis.

| Nomenclature | Factor | References | AF | % |
|:---:|:---:|:---:|:---:|:---:|
| $C_{1.1}$ | Wind speed | [13–44] | 32 | 94 |
| $C_{1.2}$ | Power Density | [23,33,45,46] | 4 | 12 |
| $C_{1.3}$ | Wind direction | [14,24] | 2 | 6 |
| $C_{1.4}$ | Effective time | [18,23,45] | 3 | 9 |
| $C_{1.5}$ | Availability data | [43] | 1 | 3 |
| $C_{1.6}$ | Turbulence | [45] | 1 | 3 |
| $C_{1.7}$ | Frost periods | [42] | 1 | 3 |
| $C_{1.8}$ | Natural disasters | [33,37,38,46] | 4 | 12 |
| $C_{1.9}$ | Air density | [13,35,40,42,43] | 5 | 15 |

### 3.1.2. Geographic Category ($C_2$)

By considering Table 2, five factors were included into this category, labeled from ($C_{2.1}$) to ($C_{2.5}$). In a similar way to the previous categorization, the results are summarized in Table 8. Two factors stand out above the rest: Slope ($C_{2.1}$) and Altitude ($C_{2.2}$). From the specific literature, 24 and 13 contributions include these factors respectively. The rest of the geographic factors became less important.

**Table 8.** Geographic Category ($C_2$)—Onshore optimal location. Quantitative analysis.

| Nomenclature | Factors | References | AF | % |
|:---:|:---:|:---:|:---:|:---:|
| $C_{2.1}$ | Slope | [13,17–20,22,24,25,28–35,37–41,43–45] | 24 | 71 |
| $C_{2.2}$ | Altitude | [13,19,24,25,29,32,35,37,39,40,42–44] | 13 | 38 |
| $C_{2.3}$ | Type of terrain | [16,17,19,21,30,40,42,45] | 8 | 24 |
| $C_{2.4}$ | Roughness | [13,20,37] | 3 | 9 |
| $C_{2.5}$ | Area | [22,29,31,43,44] | 5 | 15 |

### 3.1.3. Socio-Environmental Category ($C_3$)

Regarding Table 3, 11 factors were included in this category, from ($C_{3.1}$) to ($C_{3.11}$). Table 9 shows the results according to the specific literature. Four factors stand out from the rest: Protected areas ($C_{3.1}$), Land use ($C_{3.10}$), Flora and fauna impact ($C_{3.11}$) and Agrological capacity ($C_{3.2}$) with 22, 12, 12 and nine publications that included these factors in their study. The rest of the factors oscillate between absolute frequencies of 8 and 1 work, respectively.

**Table 9.** Socio-environmental Category ($C_3$)—Onshore wind energy. Quantitative analysis.

| Nomenclature | Factor | References | AF | % |
|:---:|:---:|:---:|:---:|:---:|
| $C_{3.1}$ | Protected areas | [13,15–18,20,22,24–26,28–31,34,39–45] | 22 | 65 |
| $C_{3.2}$ | Agrological capacity | [14,17,20,22,30,31,35,37,43] | 9 | 26 |
| $C_{3.3}$ | Visual impact | [13,17,19,23,37,42,43] | 7 | 21 |
| $C_{3.4}$ | Reduction emissions | [23,33,38] | 3 | 9 |
| $C_{3.5}$ | Stroboscopic effect | [37] | 1 | 3 |
| $C_{3.6}$ | Energy-dependence contribution | [23,33] | 2 | 6 |
| $C_{3.7}$ | Noise | [13,15,19,23,36,37,42,43] | 8 | 24 |
| $C_{3.8}$ | Population | [14,16,21,46] | 4 | 12 |
| $C_{3.9}$ | Demand electricity | [17,24,45] | 3 | 9 |
| $C_{3.10}$ | Land use | [14,17,20,21,28,32,34,35,42–44,46] | 12 | 35 |
| $C_{3.11}$ | Flora and fauna impact | [15,19,21,23,26,27,30,33,37,40,42,43] | 12 | 35 |

### 3.1.4. Location Category ($C_4$)

Table 4 shows the location category factors, including 11 factors labeled from ($C_{4.1}$) to ($C_{4.11}$). Table 10 summarizes the presence of such factors in relevant contributions. From these results, six factors exceeded 10 references—see AF column—: Distance urban areas ($C_{4.4}$)—29, Distance/availability roads ($C_{4.1}$)—26, Point of Common Coupling ($C_{4.9}$)—22, Distance transmission lines ($C_{4.3}$) and Distance airports ($C_{4.8}$) with 17 references, and Distance water resources (rivers, coast, lake) ($C_{4.11}$)—15. The rest of the factors oscillated between one and 10 publications, being minor representative of this category.

**Table 10.** Location Category ($C_4$)—Onshore wind energy. Quantitative analysis.

| Nomenclature | Factor | References | AF | % |
|---|---|---|---|---|
| $C_{4.1}$ | D. Availability roads | [16,19–22,24–34,36–45] | 26 | 76 |
| $C_{4.2}$ | D. to other wind farms | [26] | 1 | 3 |
| $C_{4.3}$ | D. transmission lines | [13,14,16,21,22,25,26,30–32,34,35,37–39,42,44] | 17 | 50 |
| $C_{4.4}$ | D. urban areas | [13–18,20–32,34–36,39–45] | 29 | 85 |
| $C_{4.5}$ | D. industrial/Military zones | [26,39] | 2 | 6 |
| $C_{4.6}$ | D.from the railway network | [13,20,25,29,30,34,35,39] | 8 | 24 |
| $C_{4.7}$ | D. to ports | [26,35] | 2 | 6 |
| $C_{4.8}$ | D. airports | [13,15,17,20–22,24–26,29,31,32,36,39,40,42,44] | 17 | 50 |
| $C_{4.9}$ | D. Point of Common Coupling (PCC) | [14,17,19,22–24,26,27,29–31,33–36,38–43,46] | 22 | 65 |
| $C_{4.10}$ | D. entertainment areas–historical | [16,17,23,26–30,34,39] | 10 | 29 |
| $C_{4.11}$ | D. water resources (rivers, coast, lake) | [14,17,20,21,24–26,28–30,34,39–41,44] | 15 | 44 |

### 3.1.5. Economic Category ($C_5$)

A set of 11 factors were included in this category, see Table 5, from ($C_{5.1}$) to ($C_{5.11}$). Of these factors, three stand out in this category: Exploitation ($C_{5.9}$), Energy put into the network ($C_{5.2}$) and Infrastructure cost ($C_{5.3}$) with 10, nine and eight works including these factors, see Table 11.

**Table 11.** Economics Category ($C_5$)—Onshore wind energy. Quantitative analysis.

| Nomenclature | Factor | References | AF | % |
|---|---|---|---|---|
| $C_{5.1}$ | Energy sale price | [13,20,24,33] | 4 | 12 |
| $C_{5.2}$ | Energy put into the network | [13,14,18,20,23–25,33,41] | 9 | 26 |
| $C_{5.3}$ | Infrastructure cost | [13,20,23,24,33,38,43,46] | 8 | 24 |
| $C_{5.4}$ | NPV | [23] | 1 | 3 |
| $C_{5.5}$ | IRR | [23] | 1 | 3 |
| $C_{5.6}$ | Payback | [23,33] | 2 | 6 |
| $C_{5.7}$ | Interest loan | [20,23] | 2 | 6 |
| $C_{5.8}$ | Installed capacity | [33] | 1 | 3 |
| $C_{5.9}$ | Exploitation | [13,17,19,20,33,38,41–43,46] | 10 | 29 |
| $C_{5.10}$ | Stability voltage | [33] | 1 | 3 |
| $C_{5.11}$ | Economic contribution | [33,43,46] | 3 | 9 |

### 3.1.6. Political Category ($C_6$)

In line with Table 6, the Political category includes three factors, labeled from ($C_{6.1}$) to ($C_{6.3}$). Table 12 shows the contribution analysis according to this category. No factor exceeded the three absolute frequency publications, oscillating their values between 2 and 3.

**Table 12.** Political Category ($C_6$)—Onshore wind energy. Quantitative analysis.

| Nomenclature | Factor | References | AF | % |
|:---:|:---:|:---:|:---:|:---:|
| $C_{6.1}$ | Incentives | [20,33] | 2 | 6 |
| $C_{6.2}$ | Taxes | [20,33] | 2 | 6 |
| $C_{6.3}$ | Policy measures | [23,33,43] | 3 | 9 |

### 3.2. Offshore Analysis. Categories and Factors

In line with the previous analysis, a similar taxonomic review corresponding to the most relevant works of offshore wind power plant optimal location was carried out by the authors. In this case, and considering the specific literature available in the database described in Section 2.1, 41 contributions published between 2008 and 2018 were selected. Between 2015 and 2018, 60.98% of the total publications were published, accounting for 25 contributions in this period. Before 2015, 16 relevant papers were published, see Figure 5. These results provide a preliminary indicator similar to the onshore analysis discussed in Section 3.1. Indeed, the evaluation and selection of optimal offshore wind power plant locations have been a remarkable interest for researchers since the 2010s, and highlighting their relevance during the recent years. In terms of the number of case studies by marine areas, the North Sea tops the list with 13 contributions, followed by the North Atlantic Ocean and the China Sea with 10 and seven studies, respectively. The continental ranking is led by Europe with 60%, followed by Asia with 33%, North America with 6% and Africa with 1%, see Figure 6. The results show that evaluation and selection of optimal locations for offshore wind power plants are mostly centralized in the northern hemisphere. The categories proposed in Section 2.2 are following discussed according to the different offshore wind power plant optimal location methodologies. With this aim, the most relevant factors taken into account in such methodologies are determined and categorized accordingly.

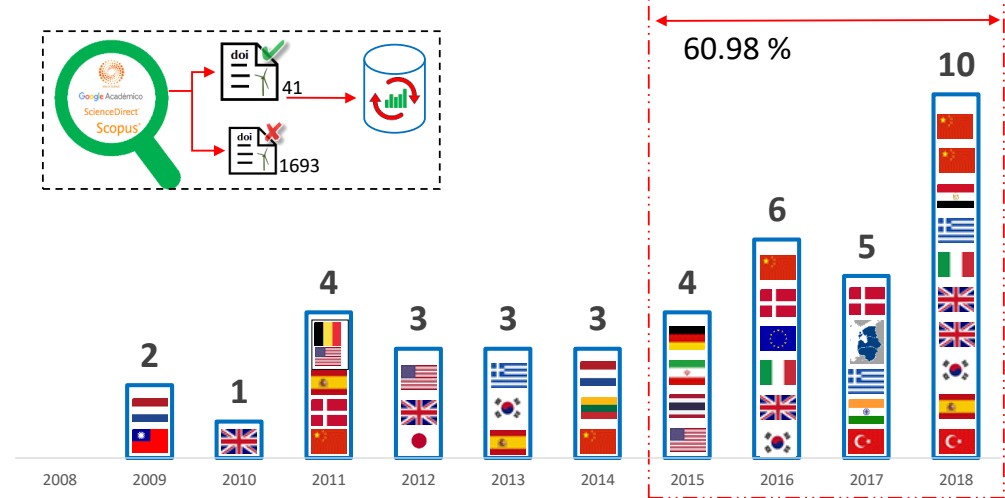

**Figure 5.** Offshore wind power plant optimal location. Frequency of publications (2008–2018).

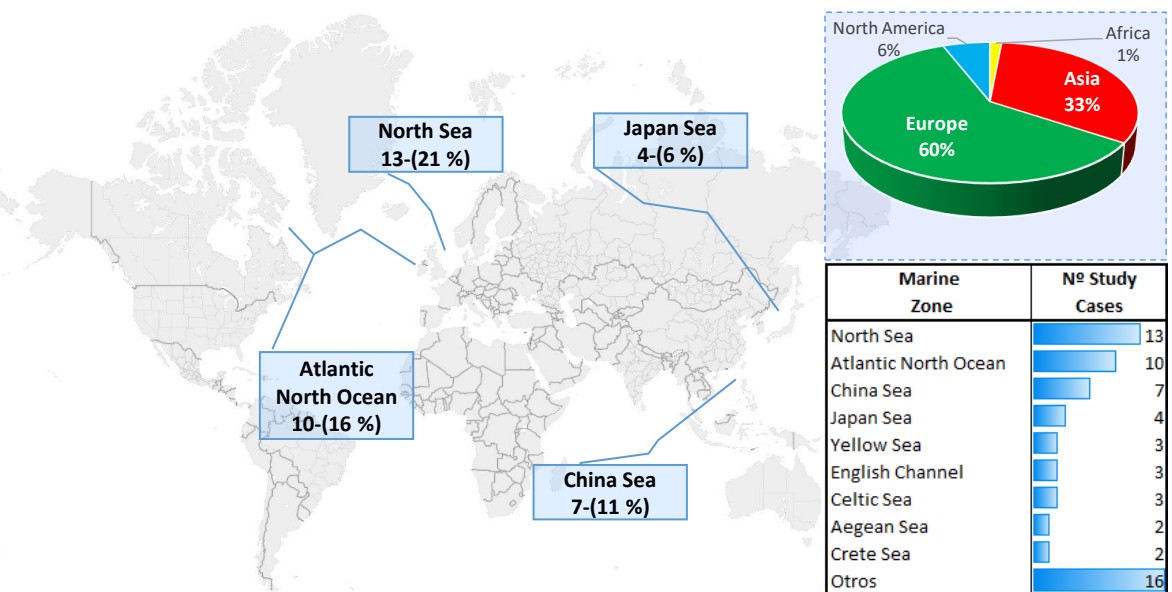

**Figure 6.** Offshore wind power plant optimal location. Geographical classification of case studies (2008–2018).

### 3.2.1. Climate Category ($C_1$)

In line with the factors included in the climate category and summarized in Table 1, seven factors were identified for offshore optimal location proposals: from ($C_{1.1}$) to ($C_{1.4}$), ($C_{1.6}$), ($C_{1.8}$) and ($C_{1.9}$). The factor with the most presence in the contributions and the highest absolute frequency was Wind Speed ($C_{1.1}$). Indeed, 37 publications directly included this factor in their proposed study. Table 13 describes the climate factors as well as the number of contributions and the percentage according to the total offshore wind power plant optimal methodologies selected.

**Table 13.** Climate Category ($C_1$)—Offshore wind energy. Quantitative analysis.

| Nomenclature | Factor | References | AF | % |
|:---:|:---:|:---:|:---:|:---:|
| $C_{1.1}$ | Wind speed | [47–83] | 37 | 91 |
| $C_{1.2}$ | Power Density | [51,54–57,65,67,77,81,84–86] | 12 | 29 |
| $C_{1.3}$ | Wind direction | [47,66,68,70,76,77] | 6 | 15 |
| $C_{1.4}$ | Effective time | [51,54,76,82] | 4 | 10 |
| $C_{1.6}$ | Turbulence | [54,68,83] | 3 | 7 |
| $C_{1.8}$ | Natural disasters | [54,55] | 2 | 5 |
| $C_{1.9}$ | Air density | [70] | 1 | 2 |

### 3.2.2. Geographic Category ($C_2$)

In this case, and according to Table 2, five factors have been used for offshore installation optimal location: ($C_{2.3}$) and from ($C_{2.5}$) to ($C_{2.8}$). One factor stood out above the rest: Water depth ($C_{2.6}$), with 24 contributions including this factor in the proposed methodology. From the rest of factors, three varied between 10 and 13 items, Type of terrain ($C_{2.3}$), Wave height ($C_{2.7}$) and Area ($C_{2.5}$) with a percentage of publications lower than 35%, see Table 14.

**Table 14.** Geographic Category ($C_2$)—Offshore wind energy. Quantitative analysis.

| Nomenclature | Factor | References | AF | % |
|---|---|---|---|---|
| $C_{2.3}$ | Type of terrain | [47,50,54,55,64,69,75,78,84,87] | 10 | 24 |
| $C_{2.5}$ | Area | [56,66,71–76,78,79,81,82,87] | 13 | 32 |
| $C_{2.6}$ | Water depth | [47,48,50–56,58–66,68,69,71,72,74–79,81–84,86] | 33 | 80 |
| $C_{2.7}$ | Wave height | [51,53–55,57,59,69,74,78,79,82] | 11 | 27 |
| $C_{2.8}$ | Water quality | [56,79] | 2 | 5 |

### 3.2.3. Socio-Environmental Category ($C_3$)

From the 11 factors initially classified in Table 3 for the Socio-environmental Category, nine factors were selected by the different contributions for offshore installation optimal location: ($C_{3.1}$) ,($C_{3.3}$), ($C_{3.4}$), ($C_{3.7}$)-($C_{3.9}$),($C_{3.11}$)-($C_{3.13}$). According to the analyzed works, four factors stand out from the rest: Protected areas ($C_{3.1}$), Shipping Routes ($C_{3.12}$), Flora and fauna impact ($C_{3.11}$) and Fishing areas ($C_{3.13}$) with 30, 28, 22 and 16 studies that included these factors in their study. The rest of the factors ranged between seven and one absolute frequency and they did not exceed a 20% of the contributions, see Table 15.

**Table 15.** Socio-environmental Category ($C_3$)—Offshore wind energy. Quantitative analysis.

| Nomenclature | Factor | References | AF | % |
|---|---|---|---|---|
| $C_{3.1}$ | Protected areas | [47,50,52,53,55–58,61–63,65,67,69,71–79,81–87] | 30 | 73 |
| $C_{3.3}$ | Visual impact | [55,62,63,73,81,85,87] | 7 | 17 |
| $C_{3.4}$ | Reduction emissions | [54,60,64] | 3 | 7 |
| $C_{3.7}$ | Noise | [53] | 1 | 2 |
| $C_{3.8}$ | Population | [80,87] | 2 | 5 |
| $C_{3.9}$ | Demand electricity | [63,71,85] | 3 | 7 |
| $C_{3.11}$ | Flora and fauna impact | [47,53–57,59–63,65,67,69,73–75,79–81,85,86] | 22 | 54 |
| $C_{3.12}$ | Shipping Routes | [47,49–51,54–58,60–63,67,69,71–76,78,80–82,84–86] | 28 | 68 |
| $C_{3.13}$ | Fishing areas | [50,51,55,56,61,62,67,69,71,72,74,76,78,80,84,86] | 16 | 39 |

### 3.2.4. Location Category ($C_4$)

Table 4 describes the factors to be considered in this category. For offshore installations, 10 factors were explicitly considered in this category: ($C_{4.2}$), ($C_{4.4}$), ($C_{4.5}$), ($C_{4.7}$)-($C_{4.10}$), ($C_{4.12}$)-($C_{4.14}$). According to the contributions, three factors had more than 15 absolute references: Distance shore ($C_{4.13}$)—26 works, Distance industrial/Military zones ($C_{4.5}$)—21 works, and Distance Point of Common Coupling ($C_{4.9}$)—18 works. The rest of the factors range between 1 and 12 absolute reference publications, see Table 16.

**Table 16.** Location Category ($C_4$)—Offshore wind energy. Quantitative analysis.

| Nomenclature | Factor | References | AF | % |
|---|---|---|---|---|
| $C_{4.2}$ | D. to other wind farms | [50,52,58,62–64,69] | 7 | 17 |
| $C_{4.4}$ | D. urban areas | [47,87] | 2 | 5 |
| $C_{4.5}$ | D. industrial/Military zones | [47,49,50,52,55,56,58,61–63,67,69,71,73–76,78,82,84,85] | 21 | 51 |
| $C_{4.7}$ | D. to ports | [47,50,51,54,56–58,61,64,77,78,84] | 12 | 29 |
| $C_{4.8}$ | D. airports | [49,71,75] | 3 | 7 |
| $C_{4.9}$ | D. Point of Common Coupling (PCC) | [48,50,52–58,61,64,67,75,78,80,82,84,86] | 18 | 44 |
| $C_{4.10}$ | D. entertainment areas—historical | [47,50,52,55,61,71,75,87] | 8 | 20 |
| $C_{4.12}$ | D. underground cables or pipes | [47,49,50,61,69,73,76,78,81,84–86] | 12 | 29 |
| $C_{4.13}$ | D. to shore | [47–50,52–62,64,69,71–73,76,77,80,83,84,87] | 26 | 63 |
| $C_{4.14}$ | D. other point | [47] | 1 | 2 |

### 3.2.5. Economic Category ($C_5$)

This category accounts for 12 factors, see Table 5. All of them were used for offshore wind power plant optimal location methodologies, which were labeled from ($C_{5.1}$) to ($C_{5.12}$). By considering the selected contributions, four factors stand out in this category: Installed capacity ($C_{5.8}$), Infrastructure cost ($C_{5.3}$), Exploitation ($C_{5.9}$) and Energy put into the network ($C_{5.2}$) with 20, 19, 18 and 11 absolute reference works. The rest of the factors have less than 10 absolute frequency contributions and they have a minor relevance in this category—less than 20% of the contributions used such factors—, see Table 17.

**Table 17.** Economics Category ($C_5$)—Offshore wind energy. Quantitative analysis.

| Nomenclature | Factor | References | AF | % |
|:---:|:---:|:---:|:---:|:---:|
| $C_{5.1}$ | Energy sale price | [54,59,71–73,82,83,85] | 8 | 20 |
| $C_{5.2}$ | Energy put into the network | [50,52–55,59,71,74,82,83,85] | 11 | 27 |
| $C_{5.3}$ | Infrastructure cost | [50,52–55,59,60,62,64,65,68,69,71–73,78,82,83,85] | 19 | 46 |
| $C_{5.4}$ | NPV | [54,67,82,83] | 4 | 10 |
| $C_{5.5}$ | IRR | [54,82] | 2 | 5 |
| $C_{5.6}$ | Payback | [54,55,82,85] | 4 | 10 |
| $C_{5.7}$ | Interest loan | [50,59,60,85] | 4 | 10 |
| $C_{5.8}$ | Installed capacity | [50–52,59,60,62,64–69,71–74,76,83,85,86] | 20 | 49 |
| $C_{5.9}$ | Exploitation | [50,52–55,59,60,62,64,65,69,71–73,78,82,83,85] | 18 | 44 |
| $C_{5.10}$ | Stability voltage | [54] | 1 | 2 |
| $C_{5.11}$ | Economic contribution | [53,58,59] | 3 | 7 |
| $C_{5.12}$ | Decommission cost | [64,68,71,82] | 4 | 10 |

### 3.2.6. Political Category ($C_6$)

Finally, and in line with Table 6, two factors were considered in this category for offshore installation optimal location: ($C_{6.1}$) and ($C_{6.3}$). These factors have less than 2 absolute reference publications, as can be seen in Table 18.

**Table 18.** Political Category ($C_6$)—Offshore wind energy. Quantitative analysis.

| Nomenclature | Factor | References | AF | % |
|:---:|:---:|:---:|:---:|:---:|
| $C_{6.1}$ | Incentives | [54,60] | 2 | 5 |
| $C_{6.3}$ | Policy measures | [53] | 1 | 2 |

### 3.3. Final Discussion

#### 3.3.1. Categories: Comparison and Statistics

Figure 7 compares the presence of the different factors—divided by categories—in the analyzed contributions for both onshore and offshore optimal location. As can be seen, the most used categories in both onshore and offshore technologies are Location ($C_2$) and Socio-environmental ($C_3$), as there are many restrictive factors associated with the building stage of such wind power plants. Offshore technology is much more expensive than onshore technology. This aspect can be deduced from the relevance of the Economic ($C_5$) category, which is the third most relevant category in offshore optimal location methodologies. The annual trend keeps similar to the general analysis evaluation by categories, being Location ($C_4$) the most common category and the Political ($C_6$) the least–used category. By considering the annual tendencies, it can be affirmed that no-pattern were deduced to estimate the presence of any factor at a specific time in the analyzed period.

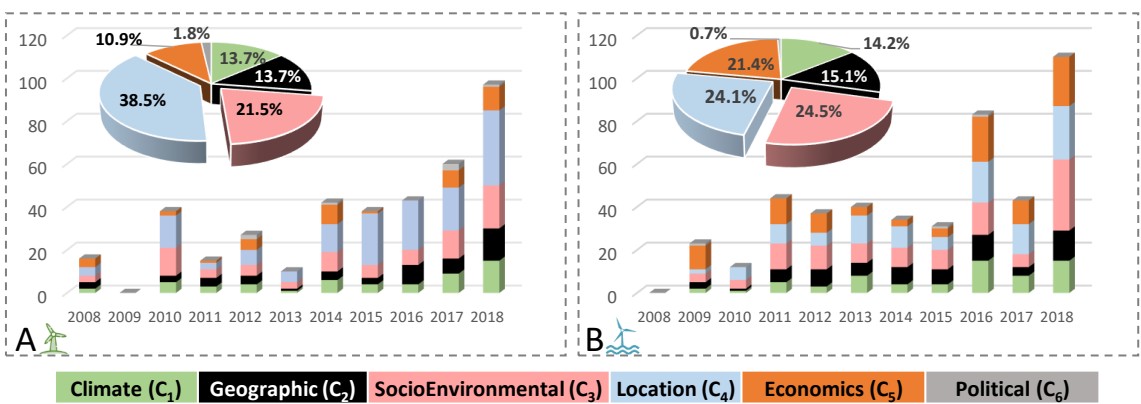

**Figure 7.** Categories: relevance and comparison (2008–2018). (**A**) Onshore wind power plant. (**B**) Offshore wind power plant.

### 3.3.2. Methodologies: Comparison and Statistics

With regard to the evaluation methods of the onshore optimal locations, and according to the specific literature, 70.6% of contributions included a combination of geospatial tools and multicriteria decision methods,—accounting for 24 works in total—. The most commonly used combination was GIS and MCDM methods, or several of them: AHP (Analytic Hierarchy Process), FAHP (Fuzzy Analytic Hierarchy Process), OWA (Ordered Weighted Average), TOPSIS, WLC (Weighted Liner Composition) [15,17,27,28,30,31,34–36,39–42,44,45]. The combination of GIS and SMCA (Spatial Multicriteria Analysis), DSS (Decision Support System), SDSS (Spatial Decision Support Systems) and MCE (Multicriteria Evaluation) can be identified in seven contributions [16,18,19,21,29,32,37]. In addition, two papers combined GIS with ELECTRE III and SMAA-TRI (Stochastic Multicriteria Acceptability Analysis) accordingly [22,25]. In terms of the optimal marine locations, it contains 63% of publications with a combination of geographic information systems and multicriteria evaluation methods (33%) or the application of only a geographic information system with an internal decision criteria (30%). The application of a GIS–MCDM combination or a combination of some of them was used in 13 studies, by considering the following processes: AHP (Analytic Hierarchy Process), FAHP (Fuzzy Analytic Hierarchy Process), OWA (Ordered Weighted Average), TOPSIS [48,52,55,57,58,63,64,67,69,71,79,80,84]. Figure 8 summarizes the methodologies proposed to estimate the optimal location for both onshore and offshore installations, as well as the relevance of each methodology according to their percentage in terms of the total contributions considered for each technology.

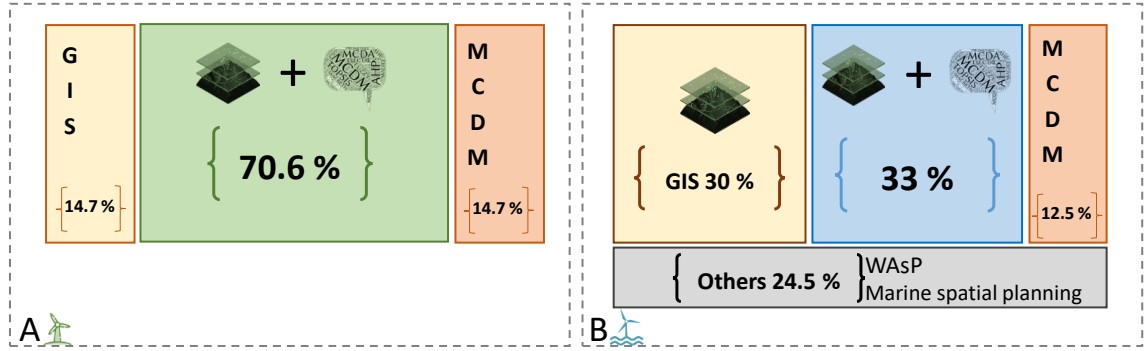

**Figure 8.** Statistics and comparison of location methodologies (2008–2018). (**A**) Onshore wind power plant. (**B**) Offshore wind power plant.

### 3.3.3. Relevant Factors: Comparison and Statistics

The most relevant factor, as expected, is Wind speed ($C_{1.1}$), directly proportional to the existence of wind power plants. Factors related to the geography of the place stand out in both technologies, such as Slope ($C_{2.1}$) and Altitude ($C_{2.2}$) in onshore plants, and Water depth ($C_{2.6}$) in offshore plants. Restrictive environmental and location factors match both technologies, such as Protected areas ($C_{3.1}$), as well as areas that are directly incompatible with this type of facilities. In addition, the Distance to Point of Common Coupling ($C_{4.9}$) is a remarkable factor due to the aim of minimizing costs and power losses. Among the determining factors of offshore optimal locations, the existence of three factors of the economic category—Infrastructure cost-CAPEX ($C_{5.3}$), Installed capacity ($C_{5.8}$) and Exploitation-OPEX ($C_{5.9}$)—and the absence of such factors in onshore optimal location processes also stands out. In addition, the Distance to shore factor ($C_{4.13}$) was proposed in 26 publications, 63% of the total offshore works. It is important to take into account the fact that most developed countries incorporate restrictive distances for industrial marine sites, and thus, some factors such as visual impact, conflicts with other industrial or tourism activities are involved. Figure 9 shows the determining factors for both technologies. The percentages provide the relative relevance of such factors for the corresponding technology, in terms of the number of contributions where each factor is considered for optimal location estimation.

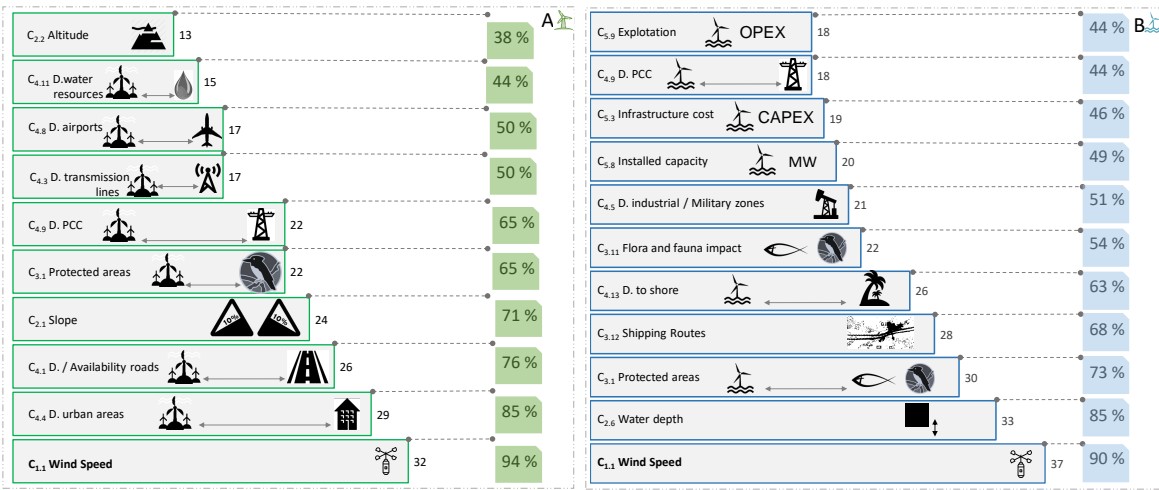

**Figure 9.** Relevant factors for optimal location methodologies. (**A**) Onshore wind power plant. (**B**) Offshore wind power plant.

Finally, Figure 10 graphically summarizes all the factors and categories proposed by the analyzed contributions included in this review. As can be seen, there are relevant factors only used by onshore case studies and vice-versa. From our point of view, the socio-environmental category ($C_3$), and more specifically Energy dependence contribution factor ($C_{3.6}$), should be considered with a higher relevance in both technologies, by considering the remarkable necessity to reduce global energy dependences. On the other hand, Factor Decommission cost ($C_{5.12}$) was only analyzed by the contributions for offshore optimal locations, although it should be considered in both technologies either in the closing of the activity or the repowering. Additionally, the existence or not of Taxes ($C_{6.2}$), belonging to the political category, is another factor to consider in both technologies, which, given the activity, should be exempted. For these reasons, and in addition to the proposed categorization, relocation of the factors, and criteria to be used in the future evaluation and selection of onshore and offshore wind optimal locations, we propose a relevant group of factors for each technology to be considered in future optimal location methodologies. These extended number of factors will allow us to estimate optimal locations from a multi-dimensional perspective, and including not only technical criteria but also environmental and energy-dependence aspects. This alternative proposal of relevant factors to be

considered in onshore and offshore optimal location methodologies is depicted in Figure 11, where the additional factors to be considered for future works are highlighted in red color.

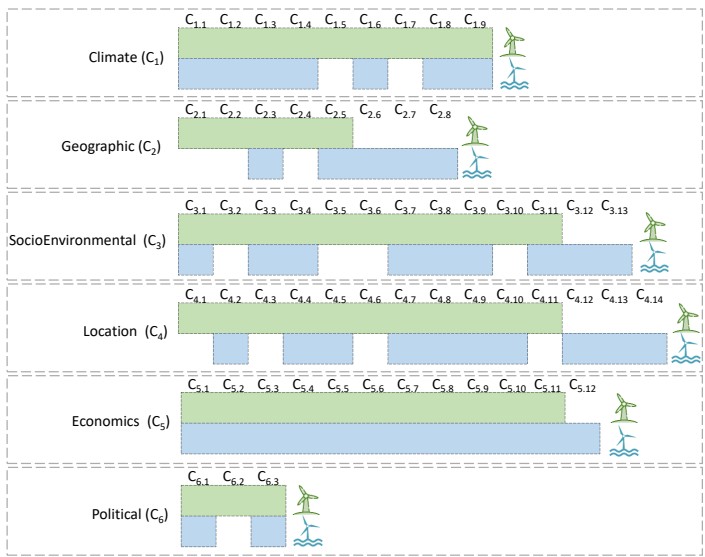

**Figure 10.** Categories and factors in onshore and offshore wind farm optimal location: global comparison.

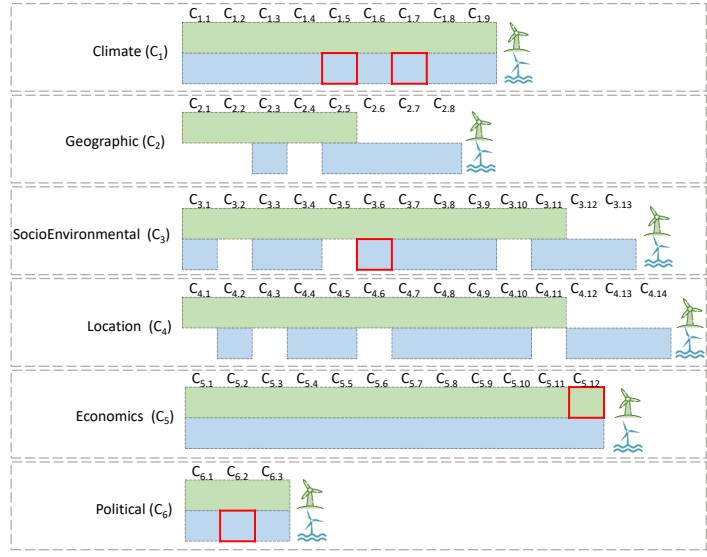

**Figure 11.** Proposal relevant factors in onshore and offshore wind farm optimal location.

## 4. Conclusions

The efficient use of renewable energy sources is extremely relevant in the global energy transition, with wind energy being the most mature technology within renewable energy sources. Each wind power plant project begins with the evaluation and selection of the optimal location. The parameters and factors to be considered for optimizing locations are different from the methodologies proposed in the specific literature. Indeed, contributions suggest different factors depending on the regulations and restrictions of each country, as well as existing data or previous studies. Under the absence of an exhaustive categorization and analysis of such factors, this paper reviews the most relevant contributions regarding the optimal location for both onshore and offshore technologies during the last

decade (2008—2018). A total of 74 contributions are identified as relevant, proposing six categories to classify a total of 59 factors: climate, geographic, economic, distance, political and socio-environmental categories. Among the all factors, the wind speed factor is considered the most relevant parameter for both onshore and offshore technologies, accounting for over 90% of the contributions. The rest of the relevant factors depend on the technology—onshore or offshore—to be implemented. Economic factors also have remarkable importance, especially in offshore wind projects where 21.4% of the contributions include some factors of this category. In terms of the methodologies proposed for optimal location estimation, 50% of researchers use the combination of GIS+MCDM in their proposed methodologies. It can be considered a very successful combination, given the multiple existing spatial data as well as the wide variety of alternatives to evaluate. By considering all factors used in optimal location methodologies, we conclude that there is a lack of environmental and energy-dependence parameters which should be included in future methodologies. An extended selection of factors is proposed by the authors. This multi-dimensional perspective will be useful for future optimal location methodology, and also it is in line with current emissions and energy-dependence reduction requirements.

**Author Contributions:** Data curation, I.C.G.-G.; methodology and formal analysis, I.C.G.-G. and M.S.G.-C.; Resources A.F.-G.; Supervision, A.M.-G.; Writing—original draft I.C.G.-G.; and Writing—review and editing, A.F.-G. and A.M.-G.

**Funding:** This research received no external funding.

**Acknowledgments:** This work was supported by 'Ministerio de Educación, Cultura y Deporte' of Spain (grant number FPU16/04282).

**Conflicts of Interest:** The authors declare no conflict of interest.

## Abbreviations

The following abbreviations are used in this manuscript:

| | |
|---|---|
| AHP | Analytic Hierarchy Process |
| DSS | Decision Support System |
| FAHP | Fuzzy Analytic Hierarchy Process |
| GIS | Geographic Information System |
| OWA | Ordered Weighted Average |
| MCDM | Multicriteria Decision-Making |
| MCE | Multicriteria Evaluation |
| PCC | Point of Common Coupling |
| PRISMA | Preferred Reporting Items for Systematic Reviews and Meta-Analyses |
| SDSS | Spatial Decision Support Systems |
| SMAA | Stochastic Multicriteria Acceptability Analysis |
| SMCA | Spatial Multicriteria Analysis |
| WLC | Weighted Liner Composition |

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
