# Peer review of "Categorization and Analysis of Relevant Factors for Optimal Locations in Onshore and Offshore Wind Power Plants: A Taxonomic Review"

_jmse, doi:10.3390/jmse7110391_

Round 1

Reviewer 1 Report

Dear editor,

I read with great interest this manuscript, which offers a comprehensive review of the factors involved in the decision process to locate suitable areas for wind power plants.

The contribution is well thought and gives insight from a source that in these cases is usually overlooked by decision makers: scientific publications on the subject.

As a review, the methodology is sound, but maybe not well articulated: it would possibly be better to simplify the language used, in order to cater to a larger audience. Some parts of the Introduction and Methodology chapters are written in a complex way that sometimes is difficult to follow. Also, it would benefit the paper to better highlight the fact this is a review in the abstract and introduction.

Also, the abstract and the Conclusions are slightly ineffective: the abstract should include the main findings of the paper and focus less on the general motivations for the study, while the conclusion should refer to wider implications without restating all the single results.

As for the findings themselves, the results are sound and align with the methodology used. Some of the implications are off: especially the claim in chapter 3.3 about the starting date for increased interest in these topics. Though publication rate is dramatically increasing, I would suggest that there is still a fair amount of time to pass from project proposals and funding to the actual year of publication. Maybe citing the exact year of the publication increase as the year of increase in interest is a bit excessive.

Graphically, I would try to reduce the number of tables and figures which is quite high: merging some of the tables would be an option. It could also be informative to merge onshore and offshore tables and figures in order to give an immediate visual comparison. Also, figures 14 and 15 are not really easy to read and would benefit a do-over.

Author Response

Dear reviewer, please, find attached our response. Thank you.

Reviewer 2 Report

The paper presents an analysis of relevant factors for optimal locations for wind power plants based on a literature review.

Manuscript’s strengths:

The review included a significant number of papers from good quality sources.

- A consistent presentation of the methodology and results.

Manuscript’s weaknesses:

The paper presents conclusions based on literature and is not necessary validated through real life experiments.

Minor recommendations for the improvement of the manuscript:

It is stated that “Finally, 70 contributions were selected to be studied.” but in Figure 2 there are presented 75. Review and correct if necessary. If Figure 3 relevant? If yes it should be better explained (for example is not clear what is “T_Soft_C” ). Also, in Figure 8, is not clear what represents the percentages. Is the occurrence in the in the 34 papers selected for analysis? How were selected that 34 out of 70 or 75 papers? Could be the windspeed considered “one of the most relevant factors for optimal location methodologies” only in 94% of cases?  In my opinion it should be relevant in all cases. In tables is ambiguous the column “AF-%AF”. I think is should be split in two columns if there is the occurrence number and the percentage of occurrence.  That minus sign could be seen as interval or something else. Not sure if “absolute frequency”  is the right term for number of occurrence. What is relative frequency? The conclusions can be extended by adding additional details regarding the findings claimed by authors.

Author Response

(The authors gave the same response as above.)
